# GroundingBooth: Grounding Text-to-Image Customization

Zhexiao Xiong[1]    Wei Xiong[2]    Jing Shi[2]    He Zhang[2]    Yizhi Song[3]    Nathan Jacobs[1]
[1] **Washington University in St. Louis**    [2] **Adobe Research**    [3] **Purdue University**

**Reviewed on OpenReview:** `https://openreview.net/forum?id=TRlZpHU3OO`

## Abstract

Recent approaches in text-to-image customization have primarily focused on preserving the identity of the input subject, but often fail to control the spatial location and size of objects. We introduce GroundingBooth, which achieves zero-shot, instance-level spatial grounding on both foreground subjects and background objects in the text-to-image customization task. Our proposed grounding module and subject-grounded cross-attention layer enable the creation of personalized images with accurate layout alignment, identity preservation, and strong text-image coherence. In addition, our model seamlessly supports personalization with multiple subjects. Our model shows strong results in both layout-guided image synthesis and text-to-image customization tasks. The project page is available at `https://groundingbooth.github.io`.

## 1 Introduction

Text-to-image customization, also known as subject-driven image synthesis or personalized text-to-image generation, is a task of generating diverse variants of a subject from a set of images with the same identity. Text-to-image customization has achieved significant progress during the past few years, allowing for more advanced image manipulation.

Earlier approaches like Dreambooth (Ruiz et al., 2023), Textual Inversion (Gal et al., 2022), and Custom Diffusion (Kumari et al., 2023) address this task by finetuning a specific model for a given subject in the test phase, which is time-consuming and not scalable. Recent approaches like ELITE (Wei et al., 2023) and InstantBooth (Shi et al., 2023) eliminate test-time-finetuning by learning a general image encoder for the subject. Although these methods improve the efficiency of inference, they mainly focus on preserving the identity of the subject, yet fail to accurately control the spatial locations of subjects and background objects. In real-world scenarios of image customization, it is a crucial user need to achieve fine-grained and accurate layout control on each of the generated objects for more flexible image manipulation.

To address this issue, in this paper, we investigate a more fundamental task, *grounded text-to-image customization*, which extends the existing text-to-image customization task by enabling spatial grounding controllability over both the foreground subjects and background objects. The input of this task includes a prompt, images of subjects, and optional bounding boxes of the subjects and background text entities. The generated image is expected to be prompt-aligned, identity preserved for the subjects, and layout-aligned for all the grounded subjects and background objects. It is challenging to satisfy all these requirements in this task simultaneously.

Several related studies have enabled layout control in text-to-image generation (Zheng et al., 2023; Li et al., 2023). However, they cannot preserve the identity of the subjects. A distinct line of research (Chen et al., 2023b; Song et al., 2022; 2024) has demonstrated control over the input subject's placement in image composition tasks. However, they are neither capable of text-to-image synthesis nor able to control the spatial location of the background objects.

To fully address our task, we propose GroundingBooth, a general framework for grounded text-to-image customization. Specifically, to enable layout control, we propose a **grounding module** that ensures both

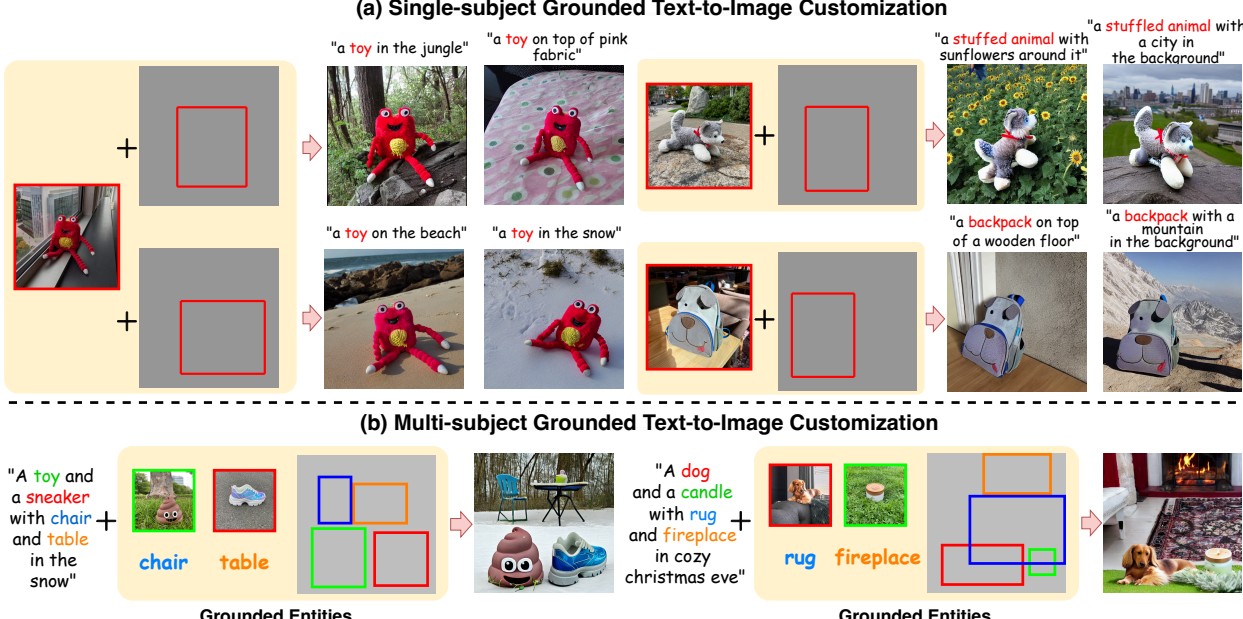

Figure 1: We propose GroundingBooth, a framework for grounded text-to-image customization. GroundingBooth supports: (a) grounded single-subject customization, and (b) joint grounded customization for multi-subjects and text entities. *GroundingBooth achieves prompt following, layout grounding for both subjects and background objects, and identity preservation of subjects simultaneously.*

the foreground subjects and background objects adhere to the input bounding boxes. Moreover, we observe that without specific design, the appearance of the generated subject tends to blended with its surrounding background objects generated from prompts Xiao et al. (2023). To resolve this issue and further improve the identity preservation of the subject, we propose a **subject-grounded cross-attention layer** that disentangles the subject-driven foreground generation and text-driven background generation, effectively preventing the erroneous blending of visual concepts. As shown in Fig. 1, our framework not only achieves grounded text-to-image customization with a single subject (Fig. 1 (a)), but also supports multi-subject customization (Fig. 1 (b)): users can input multiple subjects along with their bounding boxes, and our model can generate each subject in the exact target region with identity preservation and scene harmonization. Meanwhile, our model also allows for the grounding of multiple background objects (Fig. 1 (b)). We summarize our contributions below:

- We propose GroundingBooth, a general framework for the grounded text-to-image customization task. Our model achieves layout control for both foreground subjects and background objects while preserving subject identity. Furthermore, it supports multi-subject customization.

- We propose a subject-grounded cross-attention layer, which disentangles the foreground subject generation and text-driven background generation through cross-attention manipulation, thus preventing erroneous context blending.

- Our model outperforms existing works in text-image alignment, identity preservation, and layout alignment.

## 2   Related Work

**Text-to-Image Customization**   Text-to-image customization, also known as personalized text-to-image generation or subject-driven text-to-image generation, aims to generate images from a set of subject images

and a text prompt that describes the image content (Chen et al., 2023a; Pan et al., 2024; Xiao et al., 2023; Wang et al., 2024a; Avrahami et al., 2023). In this task, the specific identity of the input reference images is defined as a subject or a concept. Existing image customization works can be categorized into three major types. The first type is test-time-finetuning methods (Ruiz et al., 2023; Gal et al., 2022; Kumari et al., 2023). These methods tune a specific diffusion model on a few subject images so that the model is adapted to a new identifier token representing the subject. This type of methods is computationally intensive. The second type is encoder-based customization methods (Arar et al., 2023; Wei et al., 2023; Shi et al., 2023; Zhang et al., 2024), which eliminates test-time finetuning by pretraining the diffusion model equipped with an image encoder so that it can generalize to new subjects during inference. These methods can achieve much faster image customization. The third type (Roich et al., 2021; Gal et al., 2023) is a combination of the first two methods, which learns a general image encoder to encode the identity of the subject and then finetunes the model for a few steps to further improve the results. Recent works (Tan et al., 2024; Zhang et al., 2025; Wang et al., 2025) also propose using LoRA finetuning on Diffusion Transformer(DiT) architectures to achieve subject-driven customization.

Most existing image customization methods focus on synthesizing identity-preserved subject variants and are limited in controlling the layout of the generated scenes. A related work Break-A-Scene (Avrahami et al., 2023) enables personalized local region editing of an image. Their task differs from ours in that as an image editing method, they can only specify few local regions to modify, failing to fully control the layout of the full image. In contrast, our model achieves a comprehensive spatial grounding on both the foreground subjects and background contents. MS-Diffusion Wang et al. (2024b) also achieves layout control of given subjects. However, they fail to spatially control the background contents. Our model not only grounds the subjects, but also fully controls the layout of the background contents with text entities as guidance.

**Grounded Text-to-Image Generation**  Given a layout containing bounding boxes labeled with object categories or text entities, grounded text-to-image generation aims to generate the corresponding image that aligns with the layout. Traditional grounded text-to-image generation such as LostGAN (Sun & Wu, 2019), LAMA (Li et al., 2021) and PLGAN (Wang et al., 2022) are based on generative adversarial networks (GANs). Recently, diffusion-based methods (Rombach et al., 2022; Zheng et al., 2023; Li et al., 2023; Zhang et al., 2023; Wang et al., 2024c) have made attempts to add layout control for image generation. For example, LayoutDiffusion (Zheng et al., 2023) uses a patch-based fusion method. GLIGEN (Li et al., 2023) injects grounded embeddings into gated Transformer layers. ControlNet (Zhang et al., 2023) uses copied encoders and zero convolutions. InstanceDiffusion (Wang et al., 2024c) allows for multiple formats of location control. LayoutGPT (Feng et al., 2024) and LayoutLLM-T2I (Qu et al., 2023) use LLM as guidance. However, existing methods can only perform text-to-image generation without subject-driven generation and identity preservation. In contrast, our model achieves identity preservation of subjects while aligning the layout.

## 3   Our Approach

Given one or multiple background-free[1] images $\mathcal{X} = \{x_1, x_2, \cdots, x_m\}$ where each image $x_m \in \mathbb{R}^{h \times w \times 3}$ represents a subject, and their target bounding box locations $\mathcal{L}_X = \{l_X^1, l_X^2, \cdots, l_X^m\}$, text entities[2] $\mathcal{T} = \{t_1, t_2, \cdots, t_n\}$ with their target locations $\mathcal{L}_T = \{l_T^1, l_T^2, \cdots, l_T^n\}$, and the overall text prompt $\mathcal{P}$, we aim to generate a customized image $\hat{x}$, such that the subjects can be seamlessly placed inside the desired bounding box with natural poses and accurate identity, and the background objects generated from text-box pairs are positioned at the correct locations.

To rigorously distinguish the roles of these inputs, we define this task under a *Layout-Priority Paradigm*. In this formulation, the bounding boxes $(\mathcal{L}_X, \mathcal{L}_T)$ function as hard spatial constraints that define the mandatory generation regions, whereas the text prompt $\mathcal{P}$ and subject images $\mathcal{X}$ serve as semantic references that define the content. Consequently, our architecture is designed to structurally prioritize layout constraints over conflicting spatial cues embedded in the text prompt. Here $l_X^m$ or $l_T^n$ refers to the bounding box coordinates of a subject or a text entity, which can be represented as $l = [x_{\min}, y_{\min}, x_{\max}, y_{\max}]$. The generated image

---

[1]Background-free images refer to images with background removed. We obtain them with SAM (Kirillov et al., 2023).

[2]Here each text entity is referred to a text tag, such as "chair" and "hat".

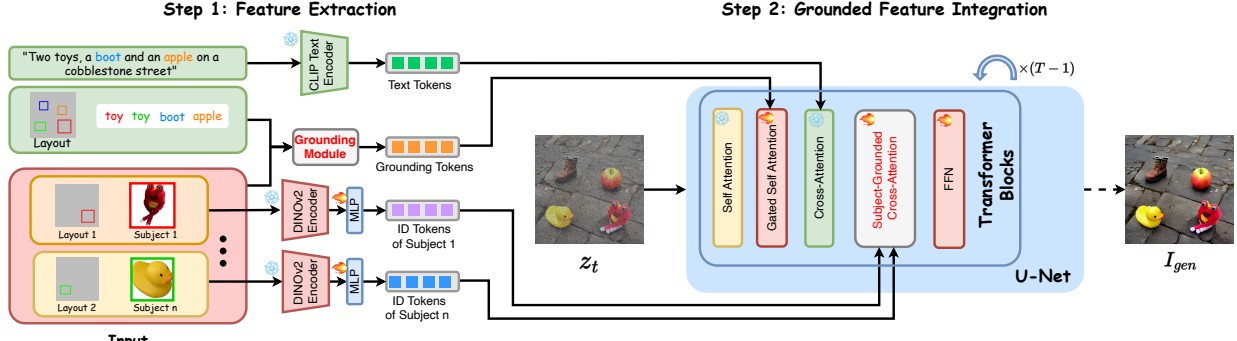

Figure 2: Inference pipeline of GroundingBooth. It contains two steps: (1) Feature extraction. We use the CLIP encoder and DINOv2 encoder to extract prompt and image tokens, respectively. We use our proposed **Grounding Module** to extract grounding tokens from layout and text entities. (2) Grounded feature integration. We propose a **Subject-Grounded Cross-Attention Layer** in each transformer block to integrate the subject image tokens, text tokens, and grounding tokens. *Note that the model is trained with a single subject per image, but generalizes well to multiple subjects during inference.*

$\hat{x}$ can be calculated as:

$$\hat{x} = \text{GroundingBooth}(\mathcal{X}, \mathcal{T}, \mathcal{P}, \mathcal{L}_X, \mathcal{L}_T). \tag{1}$$

The pipeline of our proposed GroundingBooth model is shown in Fig. 2. We first extract grounded text tokens from text and layout inputs, and image tokens from subject images, as described in Sec. 3.1. Then we integrate these tokens with our proposed subject-grounded cross-attention layer, as described in Sec. 3.2. Sec. 3.3 and Sec. 3.4 reveal the details of model training and inference, respectively.

### 3.1 Feature Extraction

**Feature Extraction of Prompt and Subject Images**  We first extract text tokens from the input prompt using the CLIP text encoder and identity tokens from the subject images using DINOv2 (Oquab et al., 2023). For each subject image, we extract 257 identity tokens which are composed of a global image class token and 256 local patch tokens. We reshape the feature dimension of each image token to 768 through a linear projection layer.

**Grounding Module**  To control the layout of the foreground and background objects, we propose a grounding module to jointly ground text and image tokens through positional encoding. Fig. 3 shows its the overall structure. Specifically, it contains two branches: 1) In the text entity branch (bottom), the bounding boxes of the background objects $\mathcal{L}_T$ are passed through a Fourier encoder to obtain the text Fourier embeddings of the text entities, which are then concatenated with the text tokens in the feature space to obtain the grounded text embeddings. 2) In the subject image branch (upper), the bounding boxes of the subject $\mathcal{L}_X$ are also passed through a Fourier encoder to extract the subject Fourier embeddings, which are then concatenated with the subject image tokens in the embedding space to obtain the grounded subject embeddings. At the end of the following two branches, the grounded text embeddings and subject image embeddings are projected via linear layers and then concatenated in the embedding space to form the final grounding tokens. Given the text entities $\mathcal{T}$ and subject images $\mathcal{X}$, the grounding process is formulated as:

$$h^{(\mathcal{T}, \mathcal{X})} = \Big[ MLP\big(\psi_{\text{text}}(\mathcal{T}), Fourier(\mathcal{L}_T)\big), \qquad MLP\big(\psi_{\text{obj}}(\mathcal{X}), Fourier(\mathcal{L}_X)\big) \Big], \tag{2}$$

where *Fourier* represents the Fourier embedding (Tancik et al., 2020), $MLP(.,.)$ is a multi-layer perceptron, $[.,.]$ is concatenation operation, and $h^{(\mathcal{T}, \mathcal{X})}$ is the grounding tokens. $\psi_{text}$ and $\psi_{obj}$ denote to the text encoder and image encoder, respectively. The generated grounding token $h^{(\mathcal{T}, \mathcal{X})}$ is an integration of the layout information of text entities, layout information of subject images, and the rich vision feature of the subject. We then inject the grounding tokens through a gated self-attention layer (Li et al., 2023) that is

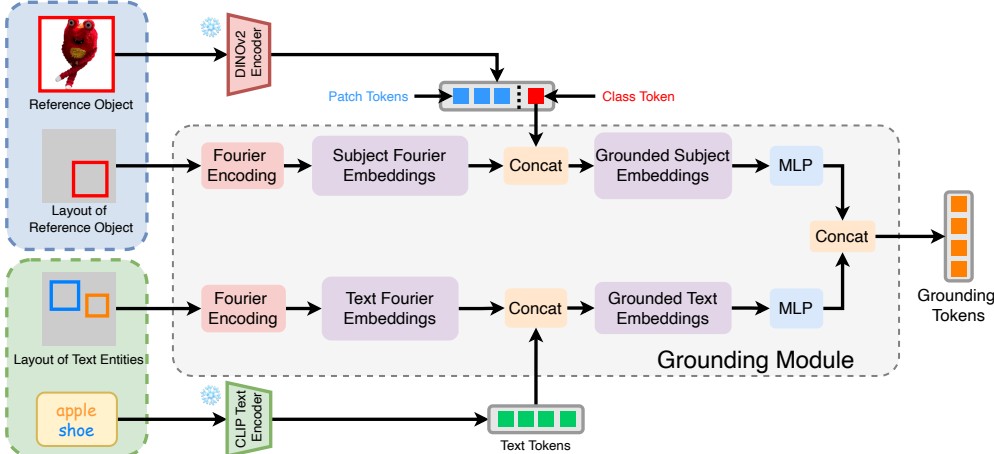

Figure 3: Grounding Module: Our grounding module takes both the prompt-layout pairs and reference object-layout pairs as input. For the foreground reference object, both CLIP text token and the DINOv2 image class token are utilized.

newly introduced into each transformer block of the diffusion U-Net, in-between the original self-attention layer and cross-attention layer of the original block. We formulate the gated attention layer as:

$$v = v + \tanh(\gamma) \cdot \left( \text{SelfAttn}\left( \left[ v, h^{(\mathcal{T}, \mathcal{X})} \right] \right) \right), \tag{3}$$

where $\gamma$ is a learnable scalar initialized as 0, $h^{(\mathcal{T}, \mathcal{X})}$ is the grounding token and $v$ is the output of the self-attention layer. During training, the model adaptively learns to adjust the weight $\gamma$ of the grounding module, which ensures stable training and balances the weight between the grounding token and the visual features.

## 3.2 Grounded Feature Integration

On fusing the text and image features, existing text-to-image customization methods usually directly concatenate the text and image tokens in the cross-attention layers, leading to several issues: First, the generated subject and the background objects generated from the prompt and text entities can be unnaturally blended, as we observe in our experiments. Second, in the circumstances where two bounding boxes belong to the same class, the model cannot distinguish whether each bounding box belongs to a subject image or a text entity, resulting in misplacement of the subject. Moreover, this type of fusion strategy usually cannot handle the customization of multiple subjects. To address all these issues, we propose a **subject-grounded cross-attention layer** to specifically disentangle the generation process of subjects and background objects. The details of our module are illustrated in Fig. 4.

**Subject-Grounded Cross-Attention Layer** In this layer, both the DINOv2 image tokens and bounding box of the subject $l_{sub}$ are taken as inputs. The queries $K$ and values $Q$ are calculated from the image tokens. We first compute the affinity matrix $A$ through $A = Q \cdot K$ and obtain $A \in \mathbb{R}^{hw \times hw}$, where $h \times w$ indicates the resolution of the feature map in the attention layer. As we have the object layout $l_{sub}$, it is straightforward to restrict the injection of image tokens only inside the region of the target bounding box. Therefore, we reshape the layout $l_{sub}$ to $h \times w$ and generate the cross-attention mask, which is formulated as:

$$M_{Layout[i,j]} = \begin{cases} 0, & [i,j] \in l_{sub}, \\ -\infty, & [i,j] \notin l_{sub} \end{cases}, \tag{4}$$

where $M_{Layout[i,j]}$ represents the mask value of position $[i,j]$ in rectified attention score maps.

The mask contains the explicit location information of the subject. It encourages the accurate placement of the subject and avoids information leakage from other objects. After obtaining the mask, we use it to

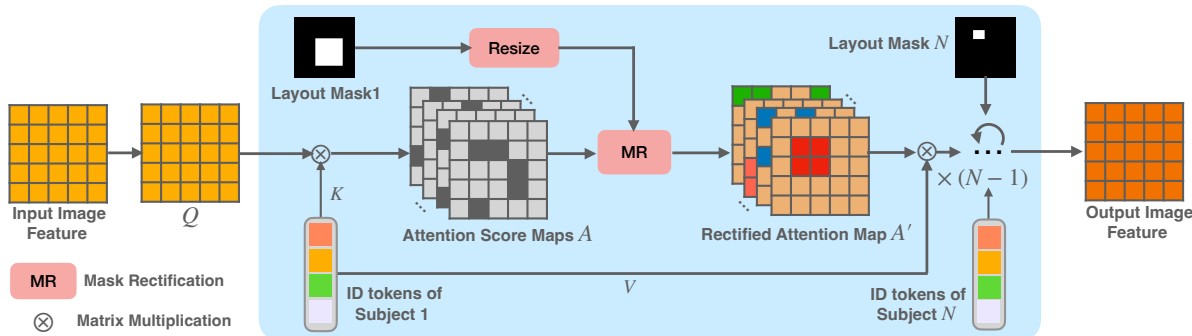

Figure 4: Subject-Grounded Cross-Attention: Q, K, and V are visual query, key, and value respectively, and A is the affinity matrix.

constrain the spatial distribution of the attention maps by rectifying the attention, and obtain the mask-rectified affinity matrix $A'$ through $A' = A + M_{Layout}$. Then we multiply the masked affine matrix $A'$ with $V$ to obtain the subject-grounded cross-attention output $f_{sub}$. The whole subject-grounded cross-attention layer is formulated as:

$$f_{sub} = \text{softmax}\left(\frac{QK^T + M_{Layout}}{\sqrt{d}}\right) V. \tag{5}$$

For the training samples where there is a lack of subject image, $M_{Layout}$ is set to all 0, then the masked cross-attention degrades into normal cross attention. Through subject-grounded cross-attention layer, the information of each subject is restricted to be integrated within the corresponding bounding box. This ensures not only the independence between the generation of foreground subjects and background objects, but also the independence among multiple subjects. Owing to this, our model seamlessly enables the customization of multiple subjects. In summary, our proposed layer prevents information leakage and ensures an accurate layout alignment of subjects.

### 3.3 Model Training

During training, for each image, we input only one subject image and its bounding box to the model, along with several text entities with their corresponding bounding boxes. The number of entities per training image is limited to 10 and we drop the rest ones. For a portion of training samples that do not contain any valid subject images or text entities, we set the token of the subject image to be zero embeddings, and the layout of the subject to be all zeros. We keep the text encoder and DINOv2 image encoder frozen and merely fine-tune the multi-layer perceptron after the image encoder, the gated self-attention layers, the subject-grounded cross-attention layers, and the multi-layer perceptron after the DINOv2 image encoder.

### 3.4 Model Inference

Although our model is trained on single-subject data, it can be seamlessly extended to achieve multi-subject customization without retraining. In the inference stage, assume we have $N$ subjects. As shown in Fig. 4, the vision token of each subject will be injected into the corresponding bounding box region via the subject-grounded cross-attention layer. As we analyzed in section 3.2, the subject-grounded cross-attention layer encourages the independence of generating each subject, preventing potential false blending of visual concepts, e.g., the unnatural blending of two objects in the overlapping regions. It also guarantees an accurate layout control on all the subjects.

## 4 Experiment

**Datasets** We mix several datasets for training. For image pairs of the same object, we use (1) multi-view data, MVImgNet (Yu et al., 2023) and (2) video instance segmentation dataset Refer-YouTube-VOS (Seo

Table 1: Comparison with existing methods on Dreambench. The top two results for each metric are highlighted with a first, second background, respectively.

| | CLIP-T ↑ | CLIP-I ↑ | DINO ↑ |
|---|---|---|---|
| BLIP-Diffusion (Li et al., 2024) | 0.2824 | 0.8894 | 0.7625 |
| ELITE (Wei et al., 2023) | 0.2461 | 0.8926 | 0.7391 |
| Kosmos-G (Pan et al., 2023) | 0.2864 | 0.8452 | 0.6933 |
| λ-eclipse (Patel et al., 2024) | 0.2767 | 0.8901 | 0.7734 |
| AnyDoor (Chen et al., 2023b) | 0.2416 | 0.9029 | **0.7781** |
| GLIGEN (Li et al., 2023) | 0.2898 | 0.8520 | 0.6890 |
| CustomNet (Yuan et al., 2023) | 0.2815 | **0.9090** | 0.7526 |
| MSDiffusion (Wang et al., 2024b) | **0.3029** | 0.8982 | 0.7267 |
| **Ours** | **0.2931** | **0.9169** | **0.7950** |

et al., 2020). MVImgNet contains 6.5 million frames from 219,188 videos across 238 object categories, with fine-grained annotations of object masks. Refer-YouTube-VOS dataset contains 3,978 high-resolution YouTube videos with 131k high-quality manual annotations and 15k language expressions. Following Any-Door (Chen et al., 2023b), for each object, we randomly selected two different frames from the same video clip to form a training pair. We apply the object mask on one frame to obtain the background-free object as the input subject image. We use the other frame as the ground-truth, and use its bounding boxes as the layout input. For single-image data, we use (3) LVIS (Gupta et al., 2019), a well-known dataset for fine-grained large vocabulary instance segmentation, including 118,287 images from 1,203 categories, and (4) OpenImages v7 dataset (Kuznetsova et al., 2020), which we only select the images with instance segmentation annotations for training. We use the ground-truth segmentation mask and crop the image to obtain the background-free subject images. For each sample of single image data, we select only the instance bounding boxes with top-10 largest areas to compose the layout, and choose the subject that has the largest area as the subject for training.

**Evaluation Metrics** We calculate the CLIP-I (Radford et al., 2021) score and DINO (Caron et al., 2021) score to assess the identity preservation performance of the subjects and use CLIP-T (Radford et al., 2021) score to evaluate the text alignment of the generated image. For evaluation of the model's grounding ability, we use $AP$ and $AP_{50}$ based on a pretrained YOLOv8 (Jocher et al., 2023) object detection model.

## 4.1 Single Subject Customization

We compare our work with existing state-of-the-art works on DreamBench (Ruiz et al., 2023) for the customization of a single subject. *We mainly compare our model with existing encoder-based customization methods, as our work falls in this line of research.* In this experiment, we use the bounding box of the subject in the ground-truth image as the input layout. The qualitative and quantitative results are shown in Fig. 5 and Table. 1, respectively. Overall, our method shows significantly better performance in layout alignment, subject identity preservation, and text alignment. BLIP-Diffusion (Li et al., 2024), ELITE (Wei et al., 2023), λ-eclipse (Patel et al., 2024) and MLLM-based method Kosmos-G (Pan et al., 2023) fail to maintain accurate identity of the subjects. They also lack the ability of precise layout control. AnyDoor (Chen et al., 2023b) is designed for image composition. It can only generate subjects on a given background, but unable to generate the background contents from texts. Although previous grounded text-to-image generation methods like GLIGEN (Li et al., 2023) can achieve layout control, it cannot preserve the identity of the subjects. CustomNet (Yuan et al., 2023) achieves pose control to some extent. However, it highly relies on the pretrained model Zero123 (Liu et al., 2023), limiting the resolution of its generated image to be $256 \times 256$. Moreover, there can be obvious artifacts around the boundary of the generated subject. MS-Diffusion (Wang et al., 2024b) can achieve grounded customization. However, it fails to maintain an accurate identity of the subject. We note that GroundingBooth and MS-Diffusion rely on different backbones, so the observed gap may reflect both the conditioning design and the underlying base model. Accordingly the results indicate different strengths: MS-Diffusion achieves higher CLIP-T on DreamBench, while GroundingBooth provides

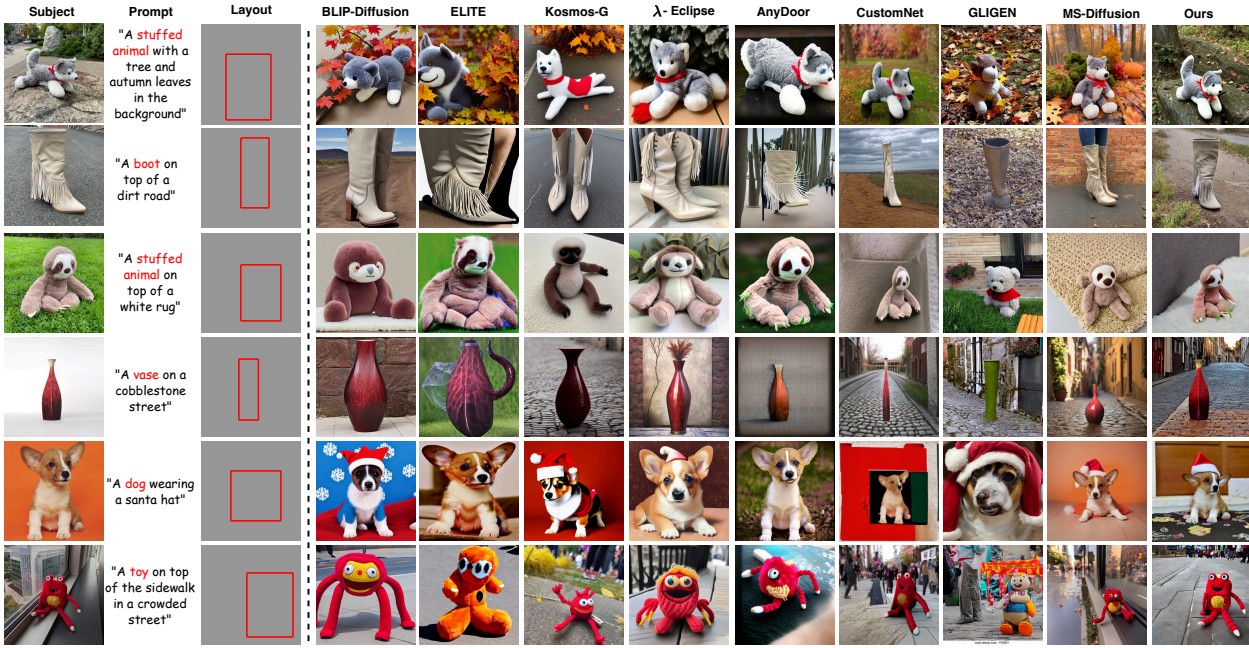

Figure 5: Visual comparison with existing methods for the single-subject customization task. Zoom in to see the details.

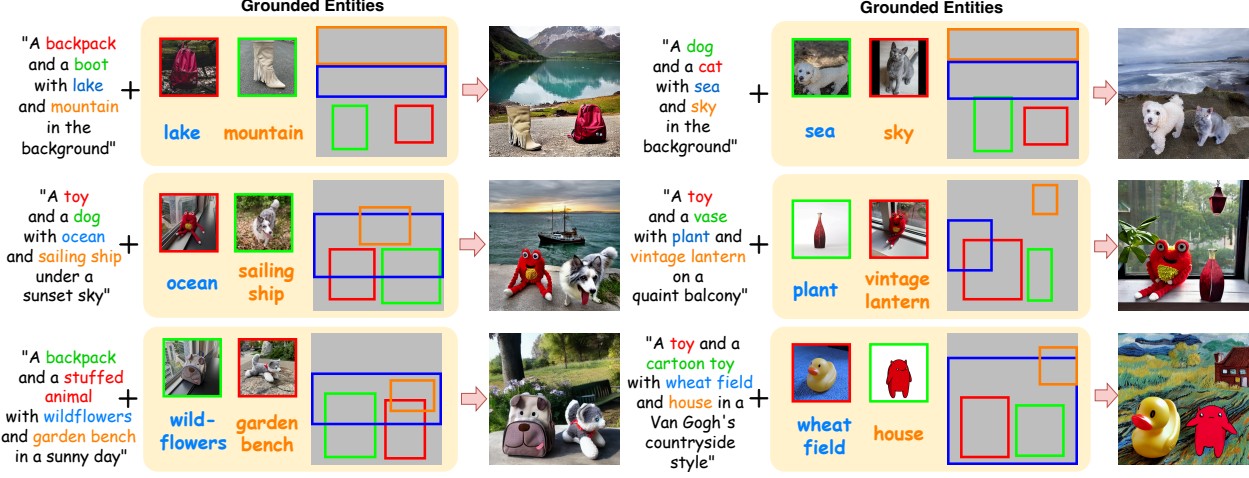

Figure 6: Multi-subject customization on DreamBench objects. Zoom in to see the details.

stronger identity preservation and more reliable joint grounding of foreground subjects and background text entities.

We observe that previous non-grounding-based customization methods tend to generate objects that are very large and in the center of the image, which increases the CLIP-I score and DINO score during evaluation. However, in real-world scenarios, users may want more control over the subject size in the generated images. They may also choose to generate a larger background with detailed textual information. In such cases, non-grounding customization methods fail to generate the desired result. Results in Fig. 5 demonstrate that our method achieves stronger identity preservation and more accurate layout alignment. We encourage the readers to view more visualizations in the Appendix.

Table 2: Quantitative results of multi-subject customization.

|  | CLIP-T ↑ | M-CLIP-I ↑ | M-DINO ↑ |
|---|---|---|---|
| λ-eclipse (Patel et al., 2024) | 0.2735 | 0.8837 | 0.7428 |
| MSDiffusion (Wang et al., 2024b) | 0.2887 | 0.8865 | 0.7153 |
| **Ours** | **0.2905** | **0.9048** | **0.7556** |

Table 3: Quantitative results of image customization with complex layout as inputs on MS-COCO validation set. In this setting, we compare our method with methods only trained on COCO.

|  | CLIP-T ↑ | CLIP-I ↑ | DINO ↑ | FID ↓ | $AP$ ↑ | $AP_{50}$ ↑ |
|---|---|---|---|---|---|---|
| LAMA (Li et al., 2021) | 0.2507 | 0.8441 | 0.7330 | 69.50 | 13.1 | 18.2 |
| UniControl (Qin et al., 2023) | 0.3143 | 0.8425 | 0.7598 | 42.22 | 4.53 | 12.8 |
| LayoutDiffusion (Zheng et al., 2023) | 0.2738 | 0.8655 | 0.8033 | 37.90 | 23.4 | 37.3 |
| GLIGEN (Li et al., 2023) | 0.2899 | 0.8688 | 0.7792 | 33.14 | 23.9 | 38.2 |
| InstanceDiffusion (Wang et al., 2024c) | 0.2914 | 0.8391 | 0.7939 | 37.57 | 36.1 | 50.3 |
| **Ours** | **0.2968** | **0.9095** | **0.8592** | **25.63** | **37.4** | **52.6** |

## 4.2 Multi-Subject Customization

With our proposed subject-grounded cross-attention layer, our model seamlessly supports the customization of multiple subjects. Fig. 6 shows the qualitative results of multi-subject customization. In this experiment, there are also multiple text entities along with their bounding boxes to describe the background contents. We observe that when inputting multiple subjects such as a bag and a boot, along with the layout of the background text entities such as the mountain and the lake, our model successfully generates the subjects and background with an accurate layout alignment for each visual concept. The identities of the subjects are preserved and the overall image is well-aligned with the prompt. Moreover, in several cases, even when the bounding boxes of the foreground objects have a large overlap with the background text entities, the model can disentangle subject-driven foreground generation from text-driven background generation, effectively avoiding context blending.

To evaluate the model's identity preservation performance on multi-subject customization quantitatively, we first compute the DINO score between each input subject and the generated image, then calculate the average score. For clarity, we name this score as Multi-DINO (M-DINO). Similarly, we follow this process but use CLIP-I score instead to obtain the Multi-CLIP-I (M-CLIP-I) score. In practice, we randomly select 2 subjects from DreamBench, and composite a layout for them as the inputs, then evaluate the models. We compare our model with baselines that support multi-subject customization. Results in Table. 2 show that our model achieves better text alignment and identity preservation in multi-subject customization.

## 4.3 Customization with Complex Layout

We evaluate our model's performance on the COCO (Lin et al., 2014) validation set, where the input layout and text entities are very complex. For each testing image, we use the largest object as the reference object (i.e., the subject), and the remaining text entities as background entities. To quantify the model's grounding ability, we adopt YOLOv8 (Jocher et al., 2023) as the object detection method, and test the evaluation results using COCO's official evaluation metrics($AP$ and $AP_{50}$). Quantitative and qualitative results are shown in Table. 3 and Fig. 7, respectively. Results show that even if we input complex layouts and text entities, our model can still generate high-quality scenes with precise layout alignment for all the objects and regions, and accurate identity preservation for the subjects, while preserving the text alignment. Compared with existing layout-to-image generation methods, our model shows a competitive accuracy in grounding the visual concepts and remarkable improvement on identity preservation.

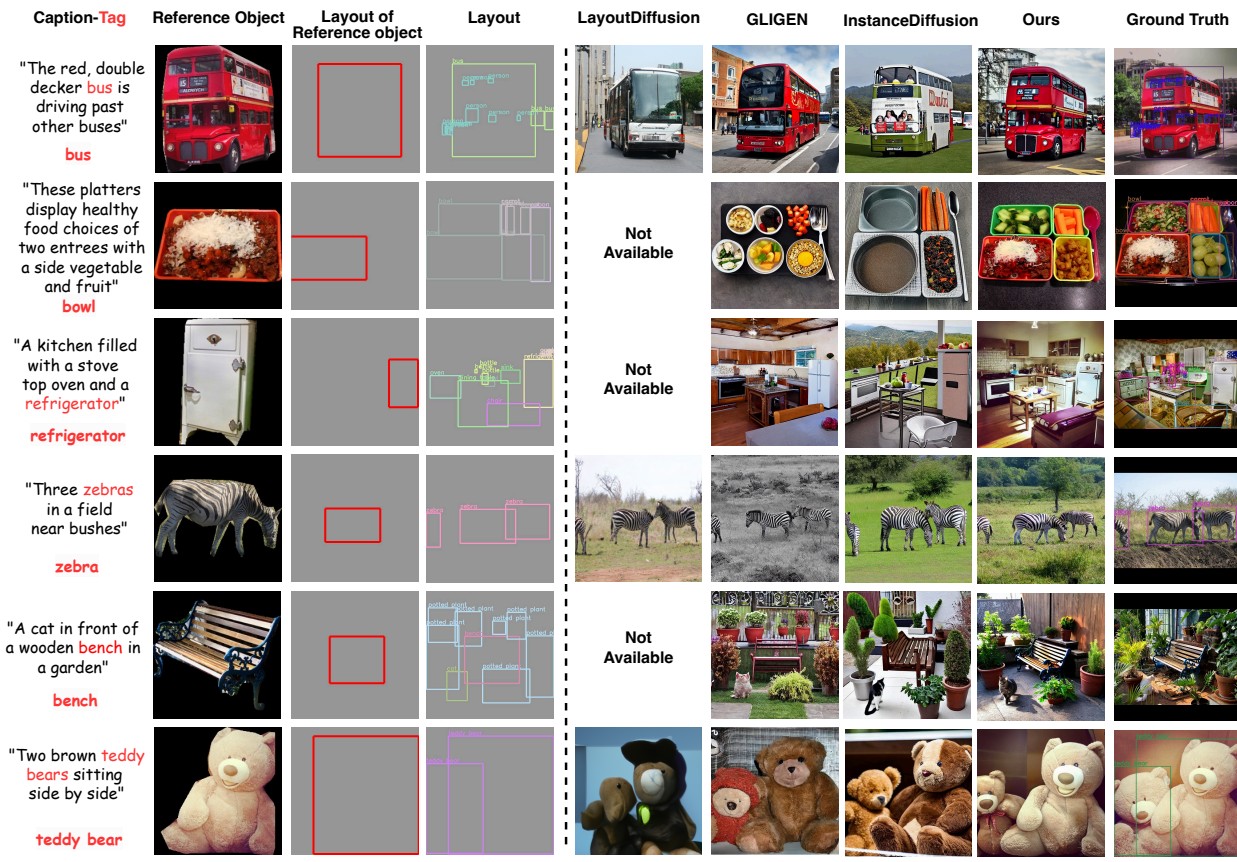

Figure 7: Visual results of image customization with complex layout and text entities as conditions on the COCO validation set. Note that LayoutDiffusion (Zheng et al., 2023) is only conducted on the COCO dataset with filtered annotations, so some of its results are not available.

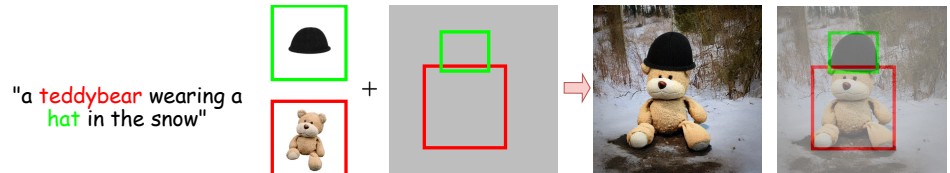

Figure 8: More results about object interactions.

## 4.4 Further Comparison

Fig. 9 provides a qualitative comparison among our method MS-Diffusion, Nano-Banana-Pro under identical prompts and layout specifications. Notice that Nano-Banana-Pro is the state-of-the-art close-source model. As shown in the examples, both MS-Diffusion and Nano-Banana-Pro struggle to consistently ground background text entities at their designated regions and exhibits noticeable spatial deviations for foreground objects, whereas our method achieves accurate grounding for both foreground reference subjects and background text entities with more reliable spatial control.

We also compare our result on VLM-based metrics in Table. 6. Specifically, we adopt VIEScore, which leverages a large vision-language model to assess the semantic alignment between the input prompt and the generated image. In our experiment, we use GPT-4o as VLM judge. Compared with previous methods, our method achieves competitive performance in prompt-following ability.

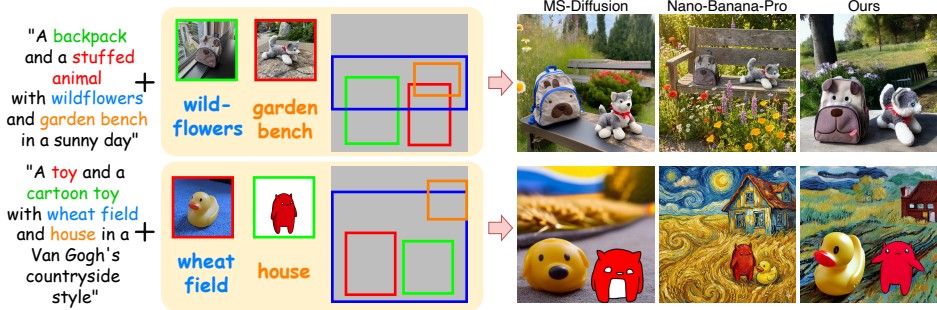

Figure 9: Qualitative comparison with MS-Diffusion and Nano-Banana-Pro. MS-Diffusion and Nano-Banana-Pro struggles to consistently ground background text entities (e.g., *wildflowers*, *garden bench*, *wheat field*, *house*) at the specified locations, and its foreground layout often exhibits noticeable spatial deviations. In contrast, our method achieves accurate grounding for both foreground reference subjects and background text entities, while providing more reliable spatial control aligned with the input bounding boxes.

Table 4: Ablation study for model components on Dreambench. GM: Grounding Module. SG-CA: Subject-Grounded Cross-Attention.

|  | CLIP-T ↑ | CLIP-I ↑ | DINO ↑ |
|---|---|---|---|
| *w/o* GM | 0.2762 | 0.8578 | 0.7049 |
| *w/o* SG-CA | 0.2878 | 0.8616 | 0.7065 |
| **Full** | **0.2931** | **0.9169** | **0.7950** |

Table 5: Ablation Study for model components on MS-COCO Validation Set. GM: Grounding Module. SG-CA: Subject-Grounded Cross-Attention.

|  | CLIP-T ↑ | CLIP-I ↑ | DINO ↑ | FID ↓ | $AP$ ↑ | $AP_{50}$↑ |
|---|---|---|---|---|---|---|
| *w/o* GM | 0.2796 | 0.8605 | 0.7740 | 40.63 | 22.1 | 28.5 |
| *w/o* SG-CA | 0.2884 | 0.8707 | 0.7970 | 34.29 | 28.5 | 38.6 |
| **Full** | **0.2968** | **0.9095** | **0.8592** | **25.63** | **37.4** | **52.6** |

## 4.5 Results on Object Interaction

Owing to the accurate layout control and identity preservation of multiple subjects, our model naturally enables realistic object interactions. Fig. 8 highlights its ability to seamlessly composite reference objects and faithfully capture their interactions.

## 4.6 Ablation Study

We conduct the ablation study to validate the effectiveness of our proposed components: the subject-grounded cross-attention layer and the grounding module. Table. 4 and Table. 5 present the quantitative results on DreamBench and COCO, respectively. We observe that both components play a vital role in improving the model's capacities of identity preservation, layout alignment, and text alignment.

In addition, our model also seamlessly support several simpler tasks by dropping some conditions, including pure text-to-image synthesis, pure layout-guided text-to-image synthesis, and the traditional personalized text-to-image synthesis. We put the related analysis and results in appendix.

We also compared our full model against a variant that replaces Fourier positional encoding with standard Linear Coordinate Encoding (i.e., directly projecting normalized bounding box coordinates via an MLP) in Table. 7. Empirical results demonstrate that substituting Fourier encoding with linear encoding leads to a noticeable degradation in layout alignment, as reflected by a decline in the YOLO-AP score. This confirms that Fourier Positional Encoding is essential for mapping positional information into a high-frequency feature space, thereby overcoming the spectral bias of neural networks and enabling the model to strictly adhere to bounding box constraints.

Table 6: Quantitative results on VLM-Based Metrics.

|  | SC ↑ | PQ ↑ | Overall ↑ |
|---|---|---|---|
| CustomNet (Yuan et al., 2023) | 7.805 | 4.472 | 5.769 |
| MSDiffusion (Wang et al., 2024b) | 7.850 | 7.256 | 7.453 |
| **Ours** | **7.891** | **7.989** | **7.690** |

Table 7: Ablation Study for different position encoding method on MS-COCO Validation Set.

|  | $AP$ ↑ | $AP_{50}$ ↑ |
|---|---|---|
| Linear | 7.1 | 11.5 |
| **Fourier** | **37.4** | **52.6** |

Table 8: User Study based on DreamBench: In the questions, the user is presented side-by-side comparisons of our generated image and another image randomly chosen from one of the baselines. The results in the table show user preference percentage.

|  | **Ours** | CustomNet | **Ours** | AnyDoor | **Ours** | GLIGEN |
|---|---|---|---|---|---|---|
| Identity | **60.78** | 39.22 | **59.31** | 40.69 | **72.81** | 27.19 |
| Grounding | **56.86** | 43.14 | **64.21** | 35.79 | **58.25** | 41.75 |
| Text Alignment | **51.96** | 48.03 | **73.52** | 26.47 | **55.34** | 44.66 |
| Overall Quality | **54.41** | 45.58 | **62.25** | 37.74 | **58.74** | 41.26 |

### 4.7 User Study

Table. 8 shows the user preference results comparing our model with existing models (Chen et al., 2023b; Yuan et al., 2023; Li et al., 2023) on DreamBench. Specifically, given the same input, we first generate results with each model. Then we ask the users to make side-by-side comparison between our result and a randomly chosen result from the baselines regarding identity preservation, text alignment, grounding ability, and overall image quality. We collect user responses using Amazon Mechanical Turk. Results show that participants have significantly higher preference over our method. We put more details in appendix.

## 5 Conclusion

We present GroundingBooth, a general framework for the grounded text-to-image customization task. Our model achieves an accurate layout grounding for both image subjects and text entities while preserving the details of the subject and maintaining text-image alignment, outperforming existing methods. Our results suggest that the proposed grounding module and the subject-grounded cross-attention layer are effective in generating distinct objects within each bounding box and improving the identity of the subjects. We hope our research can motivate the exploration of more identity-preserving and controllable foundation generative models, enabling more advanced visual editing.

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
