# Appendix

## A   Preliminary

Our model is based on Stable Diffusion v1.4 Rombach et al. (2022), a Latent Diffusion model (LDM) that applies the diffusion process in a latent space. Specifically, an input image $x$ is encoded into the latent space using a pretrained autoencoder $z = \mathcal{E}(x), \hat{x} = \mathcal{D}(z)$ (with an encoder $\mathcal{E}$ and a decoder $\mathcal{D}$). Then the denoising process is achieved by training a denoiser $\epsilon_\theta(z_t, t, f_c)$ that predicts the added noise following:

$$\min_\theta E_{z_0, \epsilon \sim \mathcal{N}(0,1), t \sim \mathrm{U}(1,T)} \|\epsilon - \varepsilon_\theta(z_t, t, f_c)\|_2^2, \tag{1}$$

where $f_c$ is the embedding of the condition (such as a prompt) and $z_t$ is the latent noise at timestamp $t$.

## B   Training/Inference Details

Our model is trained on 4 NVIDIA A100 80GB GPUs for 100k steps with a batch size of 14 and a learning rate of $5 \times 10^{-5}$ for each GPU. During training, we randomly drop reference image embedding and text embedding both at the rate of 10%. We decently rank the area of the boxes per image, and set the max number of grounding boxes to be 10 with the largest areas. During inference, we set classifier-free guidance(CFG) (Ho & Salimans, 2022) as 3.

## C   More Details of Data Collection

For each reference image, we use the segmentation mask to mask out the background and get the background-free reference object. In inference stage, we first use SAM (Kirillov et al., 2023) to get the mask of the reference object, then use the mask get the background-free reference object.

## D   More Details of User Study

Our user study is based on DreamBench (Ruiz et al., 2023), with full 30 objects and 25 prompts. We randomly generated layouts, and used them in the generation. In the user study, given the layout, the reference object, the text prompt, the result of our method and a random-selected baseline method, we request the user to answer the following four questions:

(1) Which generated image do you think that its object is more similar to the input object? Choose between Option A and B.

(2) Which generated image do you think that its object is most likely to be at the right position as the input layout? Choose between Option A and B.

(3) Which generated image do you think is most likely to match the text description? Choose between Option A and B.

(4) Which image do you think has better image quality? Choose between Option A and B.

We received more than 1200 votes from over 530 users. In the experiment, we randomly shuffle the order of baselines to improve the confidence of the user study.

## E   Additional Qualitative Results on Viewpoint Diversity

In Fig. 1 we show results about changing the shape of the bounding box. For grounded text-to-image customization, different from traditional text-to-image customization, the pose/viewpoint of the generated

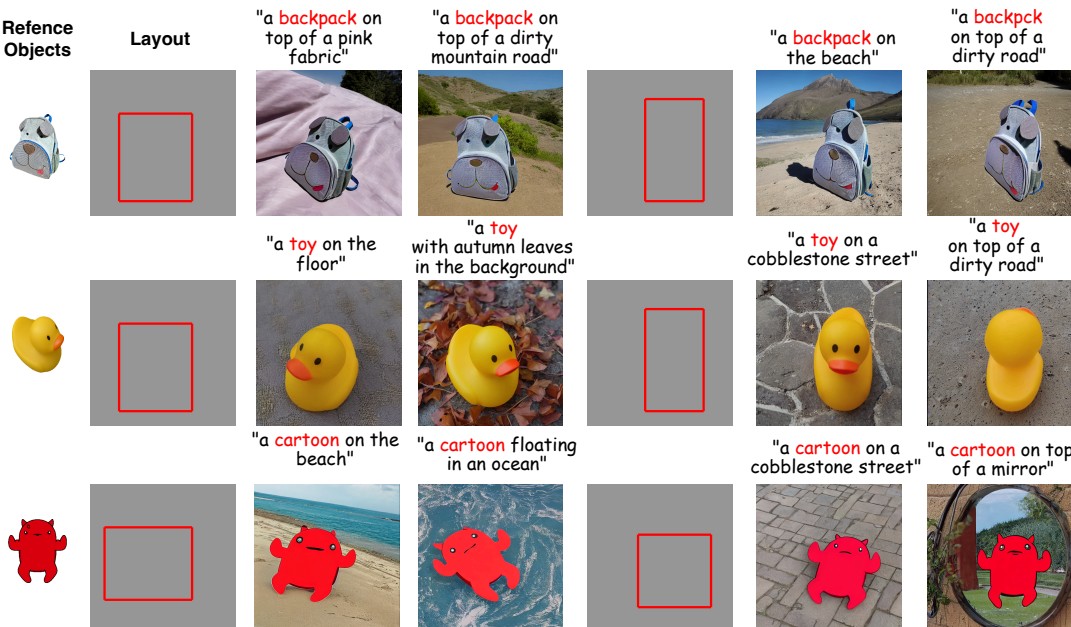

Figure 1: More visual results of our model about layout and pose change: in our model, the pose of the object is influenced by both the shape of the bounding box and the model's ability to adapt to the background. The model tends to first adapt the object into the layout, then adapt the pose to maintain harmonization with the background.

subject is jointly influenced by the shape of the bounding box and the model's ability to adapt the object to be harmonious with the background. The model tends to first adapt the object to the bounding box, then makes viewpoint adjustments to make object to be harmonious with the background. For instance, in Fig. 1, given a bounding box with a large or small width/height ratio, the grounded customized generation will generate objects with large pose change to adapt to the bounding box, then make pose refinement inside the bounding box. Users can easily conduct the initial manipulation of the object by specifying the desired layout, then the model will automatically adjust the pose of the object to be harmonious with the background. Our model shows both the ability to generate objects with accurate location and the ability to make viewpoint changes to the objects.

## F   Results on Different Grounding Conditions

Our model also seamlessly supports several simpler tasks, including pure text-to-image synthesis, pure layout-guided text-to-image synthesis, and the traditional personalized text-to-image synthesis tasks. We show qualitative results in Fig. 2 and Fig. 3.

- As shown in Fig. 2, if the bounding box is set as $[x1, y1, x2, y2] = [0, 0, 0, 0]$, the model will degrade into simpler text-to-image generation task, since the corresponding grounding tokens are set to be all-zero, and the model also loses the grounding ability.

- As shown in Fig. 3, if no reference object as input, and all the layouts rely on the input text entity to generate, then the model will degrade into layout-guided text-to-image generation task.

- If randomly assigned the bounding box of the reference object, our model is equal to the text-to-image personalization task, like previous non-grounding text-to-image customization works.

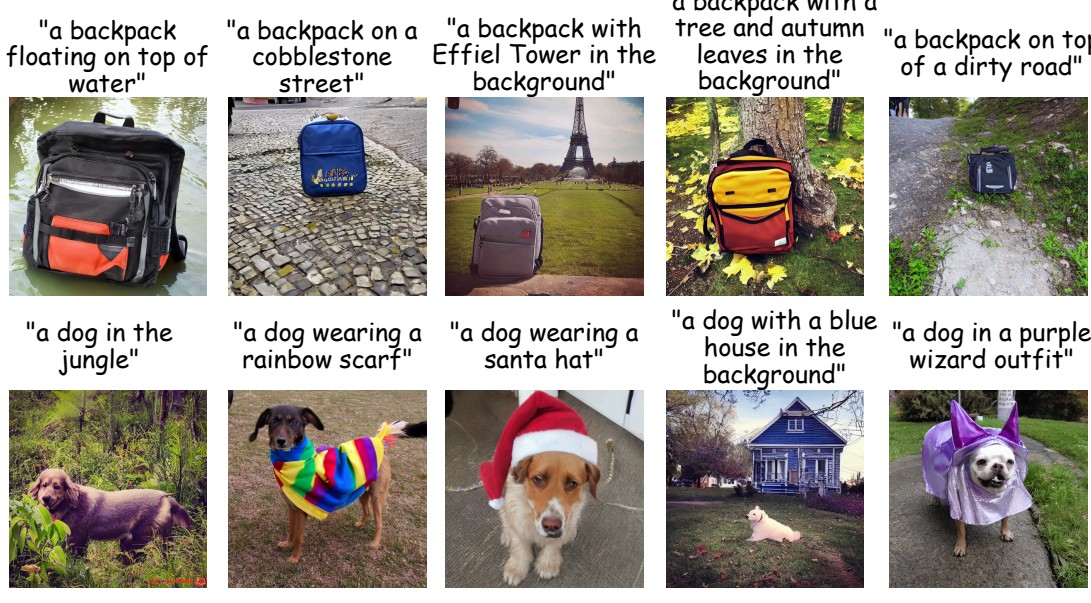

Figure 2: Our model can also deal with pure text-to-image generation task. When we set the layout $[x1, y1, x2, y2] = [0.0, 0.0, 0.0, 0.0]$, the model will degrade into a simpler text-to-image generation task, since the corresponding grounding tokens are set to be all-zero, and the model also loses the grounding ability.

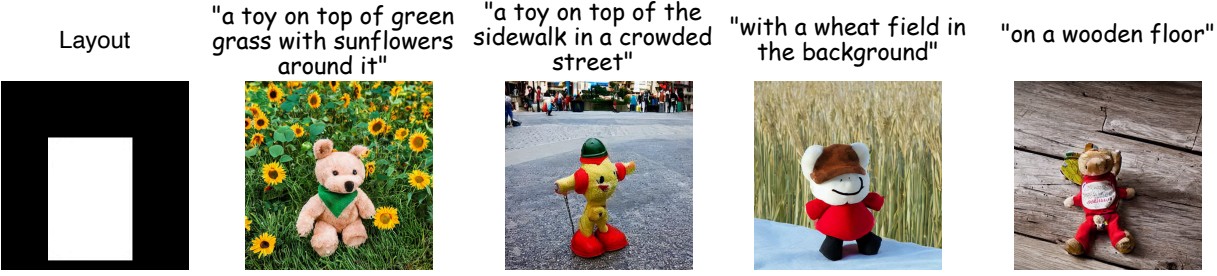

Figure 3: Our model can also deal with layout-guided text-to-image generation task: when there is no reference image input, the model will degrade into a layout-guided text-to-image generation task.

## G    More Results about pose/view change under the guidance of Prompt

We further show comparison results about pose change under the guidance of prompts in Fig. 4. We select prompts that are relevant to actions and pose change. Previous text-to-image customization models cannot maintain the identity of the reference object(row 2, row 4 and row 5), fail to achieve the prompt action-guided pose change(row 3 and row 4) and maintain text-alignment in certain cases(row 1 and row 3). Our method not only achieves grounded text-to-image customization, but is also able to maintain a good balance between identity preservation and text alignment, and can also generate objects with variations in pose.

## H    Additional Qualitative Results

In Fig. 6 we show more results about complex background evaluation on COCO2017 validation set.

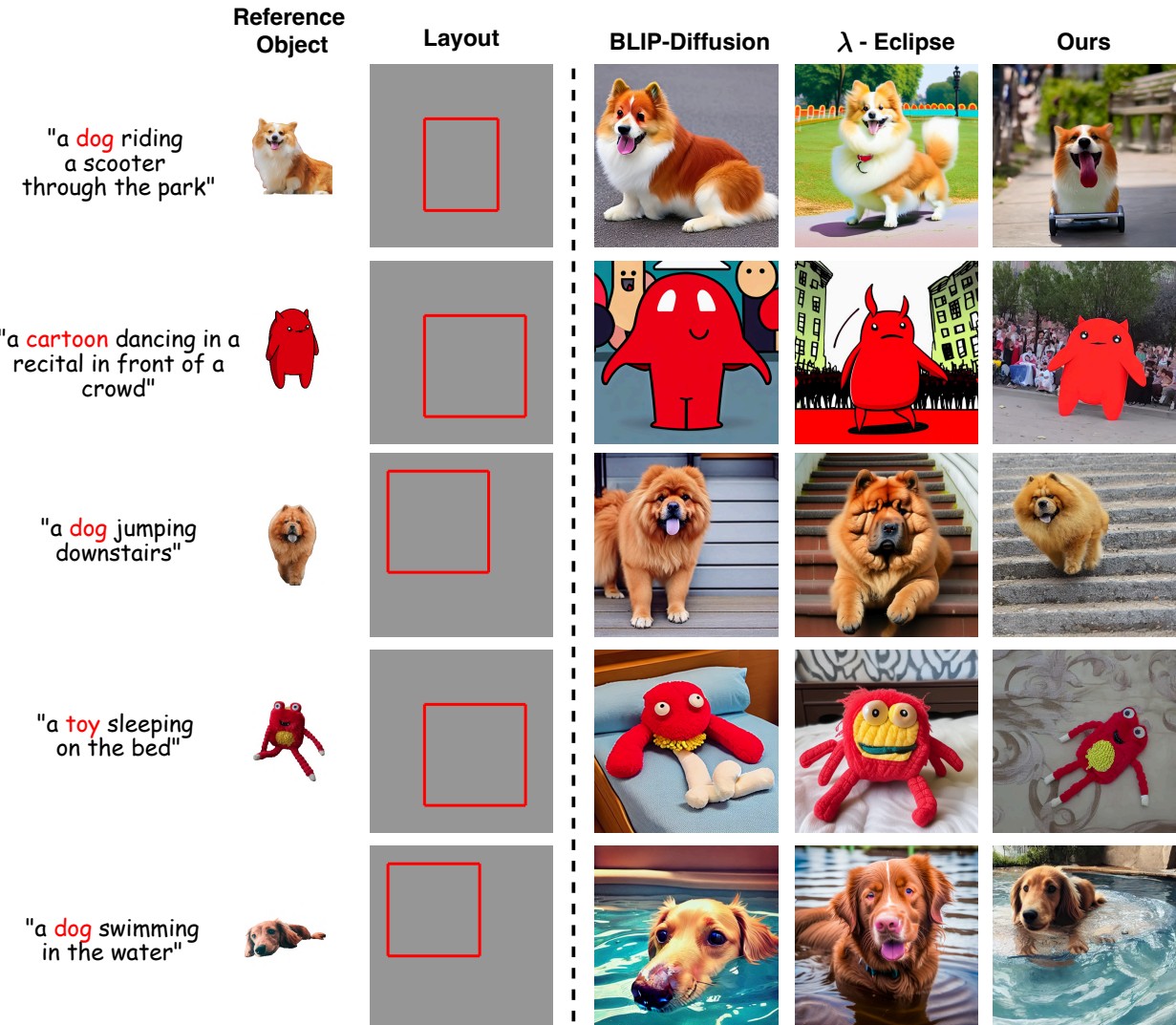

Figure 4: More results about pose/viewpoint change under the guidance of prompt.

## 1 Limitation and Future Work

Although our model successfully generates customized images with layout control, there are still several limitations. First, the model's performance can be limited by the base model. We can address this by using a stronger base model. Second, the design of injecting subject embeddings in the subject-grounded cross-attention layer in sequential could still be time-consuming during inference. This can be addressed by developing a parallel generation structure for multiple subjects. We leave these directions as future work.

While GroundingBooth achieves strong spatial grounding, it is important to note that the bounding box provides coarse geometric guidance for the subject's pose rather than explicit control. In cases where the text prompt (e.g., "standing") conflicts with the box's aspect ratio (e.g., flat), our model adheres to a layout-priority paradigm, strictly enforcing the spatial mask. This may result in poses that satisfy the geometric constraint but deviate from the textual pose description.

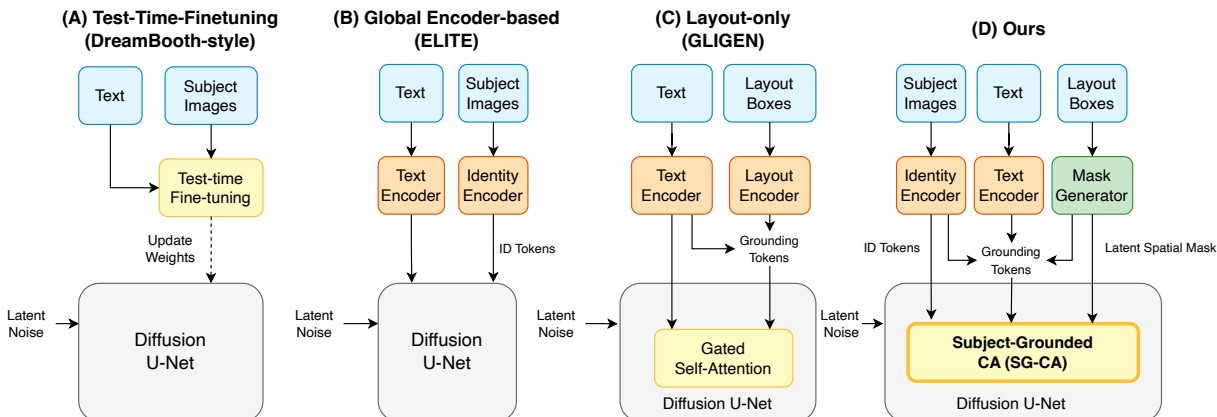

Figure 5: Model structure compare with other methods. (a) Optimization-based methods (e.g., DreamBooth) require slow, test-time fine-tuning of U-Net weights for each new subject. (b) Global Encoder-based methods encode subject images into ID Tokens and inject them globally into the U-Net. (c) Layout-only methods (e.g., GLIGEN) use Grounding Tokens derived from layout boxes and text labels to control spatial position via Gated Self-Attention, but they lack the mechanism to preserve specific subject identity from reference images. (d) GroundingBooth(Ours) achieves joint grounding and identity preservation via the novel Subject-Grounded Cross-Attention (SG-CA) module.

## J   Social Impact

GroundingBooth provides a flexible method for users to precisely customize the layout of both foreground and background objects based on user-provided reference subjects and text descriptions without any test-time finetuning. The support for the generation of multi-subjects provides a useful tool for users to generate images using their desired layout. Users can optionally choose reference objects or simple text inputs to generate their desired image, which significantly expands the flexibility in controllable and customized text-to-image generation. Nevertheless, our approach can serve as a useful tool to achieve fine-grained content creation for the AIGC community.

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

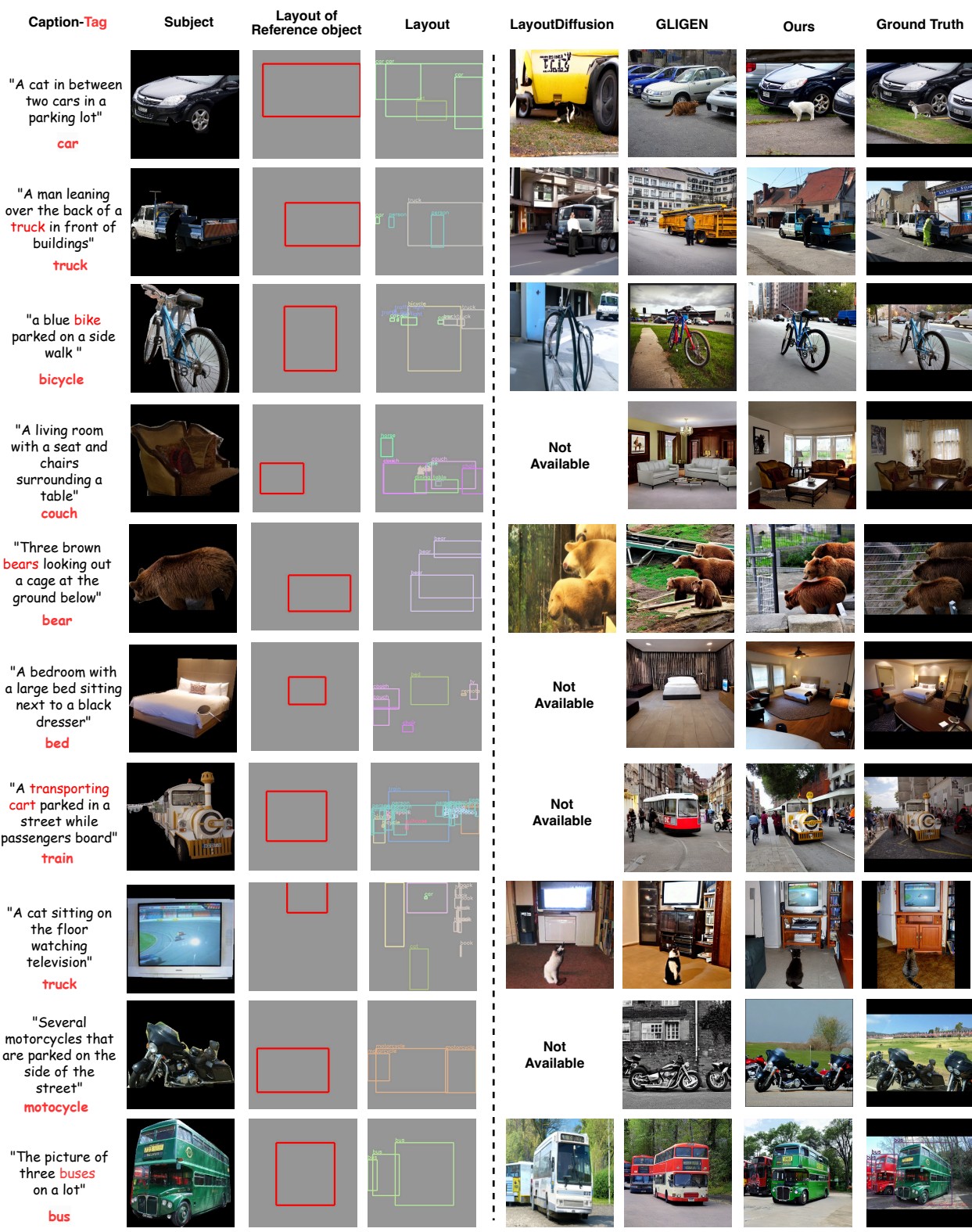

Figure 6: More results on complex scene generation on COCO validation set.