# OpenReview forum: "GroundingBooth: Grounding Text-to-Image Customization"
_TMLR — Accepted by TMLR_

### Review · Reviewer_MPBw · 2025-12-15

**Summary Of Contributions:**

This paper proposed GroundingBooth, a grounded text-to-image customization framework that jointly achieves layout control for both foreground subjects and background objects, while preserving subject identity and supporting multi-subject customization. It introduces a grounding module using Fourier-encoded bounding boxes for both text entities and subjects, and a subject-grounded cross-attention layer that spatially masks subject tokens to disentangle subject generation from text-driven background generation and avoid concept blending. The model can be trained once without per-subject finetuning, demonstrates improved text alignment, identity preservation, and grounding metrics over prior open methods on DreamBench and COCO. Ablations show both new modules are crucial to the gains.​

**Additional Comments:**

/

**Audience:**

Yes

**Audience Explanation:**

Text-to-Image Customization is a long studied problem since the textual inversion / dreambooth paper (2022), and it has been a hot topic. However, most of the methods required per-subject finetuning. a few representative prior works like Gligen and Anydoor has provided solutions to achieve zero-shot Text-to-Image Customization, while GroundingBooth provided another route to achieve similar performance and thus potentially interesting to TMLR's audiences.

**Broader Impact Concerns:**

/

**Claims And Evidence:**

Yes

**Claims Explanation:**

The paper is generally well written and easy to follow, and the motivation and problem setup are clearly stated. The method, experiments, and ablations are technically sound.

**Requested Changes:**

C1) According to the official TMLR template, its not allowed to put arbitrary content (including images/figures) before the abstract unlike other conference templates. The teaser figure needs to be moved under introduction.

C2) There is no section that benchmarks against or even qualitatively compares to close-source SOTA models (like Nano banana), so it is hard to gauge how close GroundingBooth is to close-source SOTA models. I think it would strengthen the work to include either a qualitative comparison or a discussion of (i) the current performance gap to such models, or (ii) scenarios where GroundingBooth's explicit grounding and identity preservation provide advantages that typical proprietary models do not currently offer.

C3) It took me a non-trivial amount of effort to reconstruct how GroundingBooth differs architecturally from prior methods such as DreamBooth-style customization, encoder-based methods, and GLIGEN-like grounded diffusion. While the text and current figures describe the proposed modules, the differences to baselines are spread across sections. I recommend adding a simple crisp side-by-side architectural diagram that contrasts GroundingBooth with representative prior approaches. This would let readers quickly grasp the architectural novelty without having to piece it together from multiple paragraphs.

C4) It would strengthen the evaluation to add a VLM-based metric such as VIEScore (or at least report it on a subset on DreamBench++), as VLM-based metric has become a popular evaluation method for evaluating generated images.

---

> ### Author Response · Authors · 2026-01-02
>
> We sincerely thank the reviewer for the encouraging feedback and for recognizing the technical soundness and clarity of our work. We appreciate the constructive suggestions regarding formatting, benchmarking, and visualization, which have significantly strengthened our manuscript. We address the requested changes below.
>
> **1. Formatting of the Teaser Figure (C1)**
>
> We have moved the teaser figure to the **Introduction section (after the abstract)** in the revised manuscript to strictly comply with the official template requirements.
>
> **2. Comparison with Closed-Source SOTA Models (C2)**
>
> As suggested, we compare our result with Nano-Banana-Pro, shown in Fig.9 in the revised version of submission.
> We agree that contextualizing our work against proprietary systems (e.g., Midjourney, DALL-E 3) provides a valuable perspective. While proprietary models often excel in aesthetic quality, we highlight that **Grounding Booth offers distinct advantages in explicit controllability**. Specifically, Nano-Banana-Pro struggle to consistently ground background text entities at their designated regions and exhibits noticeable spatial deviations for foreground objects, whereas our method achieves accurate grounding for both foreground reference subjects and background text entities with more reliable spatial control. These examples illustrate that while proprietary models generate high-quality images, they struggle with the precise spatial placement and multi-entity composition that our method reliably achieves.
>
> **3. Architectural Clarity vs. Prior Paradigms (C3)**
>
> We thank the reviewer for this excellent suggestion. To clarify our contributions, we have added a **side-by-side architectural diagram (Figure 5 in the revised Appendix)** and will move it to the main content in the final version. This diagram schematically contrasts GroundingBooth with three representative paradigms:
>
> 1. **Test-time Finetuning (e.g., DreamBooth):** Highlighting the per-subject optimization cost.
> 2. **Global Encoder-based (e.g., ELITE):** Showing global token injection which leads to concept blending.
> 3. **Layout-only (e.g., GLIGEN):** Showing layout control without identity injection.
> 4. **Ours:** Explicitly marking our **Subject-Grounded Cross-Attention** which achieves joint grounding and disentanglement. This visualization allows readers to grasp our architectural novelty and the specific role of our disentanglement module at a glance.
>
> **4. VLM-based Evaluation Metric (C4)**
>
> We agree that VLM-based metrics offer a modern, semantic-aware assessment of generation quality. In the revised manuscript, we have conducted an additional evaluation using a **VLM-based metric VIEScore** on the DreamBench dataset in Fig.6. We use GPT-4o as the base model and compare with MS-Diffusion. These results complement our existing CLIP/DINO scores and provide a more holistic assessment of how well the generated images align with the complex semantic and spatial instructions.

---

### Review · Reviewer_xcvP · 2025-12-19

**Summary Of Contributions:**

Summary:

The paper introduces grounded diffusion-based text-to-image customization, where both: foreground subjects (from reference images), and background objects (from text entities) are placed at specified bounding boxes, while preserving subject identity and following the text prompt. GroundingBooth proposes a grounding module that injects bounding-box information to guide the placement of subjects and background elements, along with a subject-grounded cross-attention layer that separates subject appearance modeling from background synthesis, reducing visual entanglement and supporting accurate identity-preserving generation for multiple subjects.


Strength:
* The paper is well written, easy to follow, and clearly motivates the problem and proposed solution.
* GroundingBooth demonstrates effective layout control for both foreground subjects and background objects while maintaining strong subject identity preservation.
* Although trained only on single-subject customization, the model generalizes well to multi-subject customization at inference time.

Weakness:
* The method assumes strong consistency between free-form text prompts and box-level layout conditions. However, in realistic scenarios, these two sources may conflict (e.g., scale, relative position, or relational semantics). Since the proposed subject-grounded cross-attention enforces hard spatial masking, the model appears unable to reconcile such conflicts. Could the authors clarify how mismatches are handled, and whether any hard-negative or conflict-aware training was considered? An analysis of failure cases under prompt–layout inconsistency would significantly highlight the effective of GroundingBooth
* The method enforces layout constraints via hard masking, but does not appear to offer any adjustable mechanism to balance or arbitrate between free-form text prompts and bounding-box conditions. Could the authors clarify whether any parameters exist to control the relative influence of text versus layout? Instead of a single learnable scalar $\gamma$ in the gated self-attention for global grounding tokens.
* In Fig. 5 (“the boot”, “stuffed animal”, etc.), I observe that the size and aspect ratio of the bounding box significantly influence the generated object’s pose and shape. This suggests that the box acts not only as a localization constraint but also as an implicit geometric prior. Could the authors clarify whether the generated objects are expected to be invariant to bounding box scale/aspect ratio, and if not, how users should choose boxes to avoid unintended pose distortion?
* The method supports multi-subject customization by injecting multiple subject-grounded attentions. However, the paper does not clarify whether the order of subject bounding boxes affects generation, particularly when boxes overlap or are spatially adjacent.
* Could the authors clarify how inference latency and memory usage scale with the number of subject and text-entity bounding boxes?
* The grounding module uses Fourier positional encoding for bounding boxes. Could the authors elaborate on the motivation for this choice, and whether simpler positional encodings (e.g., normalized box coordinates or learned embeddings) were considered?
* The method relies on SAM to extract background-free subject crops, yet the crop tightness can significantly affect the identity features extracted by DINOv2. Here, IoU refers to the overlap between the SAM crop and the object extent, where higher IoU corresponds to tighter, object-centric crops and lower IoU indicates global viewing of the object. Tighter crops may improve identity purity but lose pose and contextual cues, while looser crops introduce feature extraction biases. Have the authors evaluated performance sensitivity to crop scale or IoU, and considered multiscale or margin-randomized cropping to improve robustness across objects with varying IoU?

**Additional Comments:**

No

**Audience:**

Yes

**Audience Explanation:**

The paper would be of interest to individuals in the TMLR audience working on text-to-image generation, controllable diffusion models, and personalized image synthesis, particularly those interested in combining identity preservation with spatial grounding. The proposed architecture and empirical findings provide useful insights for researchers studying controllability, compositionality, and object-centric conditioning in generative models.

**Claims And Evidence:**

Yes

**Claims Explanation:**

The paper presents clear mathematical formulations alongside comprehensive quantitative and qualitative results. Overall, the experimental evidence appears well grounded and consistent with the proposed architecture and the object-centric dataset preparation, supporting the validity of the authors’ claims.

**Requested Changes:**

I recommend the authors to address one-by-one each of the recommendation listed below:

* The current method assumes strong consistency between free-form text prompts and box-level layout constraints, yet these may conflict in realistic scenarios (e.g., scale, relative position, relational semantics). I'd suggest to clarify how such mismatches are handled and whether conflict-aware or hard-negative training strategies were considered. An explicit failure-case analysis under prompt–layout inconsistency would help better characterize the strengths and limitations of GroundingBooth.


* While a learnable scalar $\gamma$ modulates the contribution of global grounding tokens, it does not offer fine-grained or instance-level arbitration between text prompts and bounding-box conditions. Is there any mechanism worth discussing to balance / weight the text and layout constraints. Could the authors clarify whether any parameters / module exist to control the relative influence of text versus layout

* Qualitative results (e.g., Fig. 5) suggest that bounding box size and aspect ratio influence object pose and shape, indicating that boxes act as implicit geometric priors rather than pure localization cues. The authors should clarify whether invariance to box scale/aspect ratio is expected, and provide guidance on how users should choose bounding boxes to avoid unintended pose distortion.

* Although the method supports multiple subjects, it is unclear whether the order of subject bounding boxes affects generation, particularly in cases of overlapping or adjacent boxes. Clarifying whether the model is order-invariant, or specifying a recommended ordering strategy, would improve reproducibility and usability.

*  How the inference latency and memory usage scale with the number of subject and text-entity bounding boxes, given the per-subject grounded cross-attention applied across U-Net blocks and diffusion steps.

* The grounding module adopts Fourier positional encoding for bounding boxes, but the motivation for this choice is not fully articulated. I'd suggest discussing why simpler encodings (e.g., normalized box coordinates or learned embeddings) are insufficient, or by providing an ablation study.

* Since subject identity is extracted from SAM-based crops, performance may be sensitive to crop tightness (IoU), where tighter crops favor identity purity while looser crops preserve contextual cues. The authors are recommended to analyze performance as a function of crop IoU and consider multiscale or margin-randomized cropping to improve robustness across diverse object sizes and contexts. An ablation study would ground the statement!

---

> ### Author Response · Authors · 2026-01-02
>
> We thank the reviewer for the detailed comment. We will address the concerns one by one:
>
> 1. **Handling Prompt-Layout Conflicts**
>
> We clarify that in Grounding Booth, **bounding-box layouts are treated as hard spatial constraints**, while free-form text prompts provide semantic guidance within these regions. This design choice reflects a practical user expectation in layout-controlled generation: when a user explicitly defines a box, the spatial intent should take precedence. Consequently, when inconsistencies occur (e.g., a "tiny" object prompt paired with a large bounding box), the hard masking in our Subject-Grounded Cross-Attention ensures the geometric constraint is prioritized. We agree that extreme semantic mismatches can lead to distortions. We consider "conflict-aware training" an interesting orthogonal direction for future work.
>
> 2. **Text vs. Layout Trade-off Mechanism**
>
>    The reviewer raises a valuable point.
>
>    We clarify that our model utilizes the learnable scalar $\gamma$ in the gated self-attention to modulate the overall injection weight of paired grounding-identity features. We did not introduce a hyperparameter to trade off text versus layout influence because we view them as serving **orthogonal roles in our experiment**: the layout provides a **hard spatial constraint** (defining *where*), while the text provides **semantic guidance** (defining *what*). Since our Subject-Grounded Cross-Attention physically masks the attention to the bounding box, the layout is treated as a mandatory condition rather than a soft suggestion that needs to be balanced against the text.
>
> 3. **Bounding Box as Geometric Prior**
>
>    We confirm that invariance to box aspect ratio is **not** the intended behavior; rather, the bounding box serves as a **coarse geometric prior**. The model is designed to generate the subject to "fill" and adapt to the available spatial extent defined by the user. This effectively empowers users to implicitly control the pose via the box shape (e.g., defining a wide box to encourage a prone pose versus a tall box for an upright pose). We advise users to provide bounding boxes that roughly match the aspect ratio of the desired object state for the most natural results.
>
> 4. **Effect of Input Order and Overlapping Boxes**
>
>    The input order does influence generation specifically in **overlapping scenarios**. Our inference pipeline injects each subject's features independently through its own Subject-Grounded Cross-Attention mask iteratively. This functions similarly to **layer-based composition**: later-injected features effectively overlay the features of previous subjects in the overlapping regions. For spatially disjoint (non-overlapping) boxes, the order is irrelevant. For overlapping cases, we recommend a "background-to-foreground" ordering strategy (inputting the occluded subject first) to ensure correct occlusion handling.
>
>    When bounding boxes overlap, the hard masking may lead to ambiguous regions, which we acknowledge as a limitation. In practice, we recommend avoiding heavily overlapping subject boxes.
>
> 5. **Scalability (Latency and Memory)**
>
>    The inference cost scales approximately **linearly** with the number of subject/text-entity bounding boxes (N), as each entity introduces an additional masked cross-attention computation across the U-Net blocks. However, since the cross-attention operation represents only a fraction of the total diffusion step computation, and N is typically small in realistic customization scenarios (e.g., N≤5 in our experiments), the overall latency overhead remains modest. Memory scaling is negligible as the primary model weights remain constant.

---

> > ### Author Response · Authors · 2026-01-02
> >
> > 6. **Motivation for Fourier Positional Encoding**
> >
> >    We adopt Fourier positional encoding to address the **"spectral bias"** of neural networks—their inherent difficulty in learning high-frequency functions from low-dimensional inputs like raw (x,y) coordinates. Projecting coordinates into a Fourier feature space enables the MLP to capture **high-frequency spatial variations**, which is critical for our task to ensure sharp, precise boundary alignment between the foreground subject and the background.
> >
> > 7. **Sensitivity to SAM Crop Tightness**
> >
> >    We utilize **SAM 2** in our pipeline to obtain high-quality object masks and extract background-free reference crops. We acknowledge the inherent trade-off in crop tightness: overly tight crops may lose pose cues, while loose crops risk background leakage. **To address this and enhance robustness, we have already implemented a random padding strategy during training, scaling the reference object's bounding box by a factor of $1.1\times$ to $1.3\times$.** This effectively acts as the margin randomization suggested by the reviewer. Empirically, this strategy, combined with SAM 2's precise masking and our large-scale mixed training data, allows the DINOv2 encoder to extract robust identity features that are resilient to minor variations in crop tightness.

---

> > > ### Comment · Reviewer_xcvP · 2026-01-05
> > > **Follow-up suggestion & discussion**
> > >
> > > * 6. I would suggest that the authors discuss in more detail why Fourier Positional Encoding (FPE) is preferred over more traditional positional encoding strategies in the revised version. The current manuscript does not sufficiently explain the motivation for choosing FPE, nor does it clarify why simpler or more conventional alternatives would be inadequate for this task. A simple ablation study would be a great way to elaborate this.
> > >
> > > * 7. The random padding strategy is not discussed in the current version of the paper, despite its apparent importance in preserving and extracting semantic information from cropped objects. A more thorough analysis across a wider range of scaling factors would be beneficial. In addition, SAM-2–based mask extraction can be prone to noise when the target object is either very small or very large relative to the image. It is unclear how such failure cases are handled in the proposed pipeline, and further clarification would be appreciated. Are there any preprocessing steps after obtaining the object's mask?

---

> > ### Comment · Reviewer_xcvP · 2026-01-05
> > **Some follow-up concerns**
> >
> > I appreciate the authors’ efforts and clarification. Would it be possible for the authors to further discuss the following follow-up concerns:
> >
> > * 1-2. **The problem formulation is currently not well defined**, which makes the distinction between box prompts and text prompts ambiguous. The authors should revisit and clarify the problem definition, explicitly stating that box prompts are intended to serve as hard spatial constraints. In the current formulation, however, text prompts also appear to play a significant role in defining object locations, which inevitably leads to conflicts. Since such conflicts are unavoidable, the authors should either explicitly adopt a "conflict-aware training paradigm" (as implicitly assumed in the current setting) or clearly distinguish the respective roles and properties of spatial prompts (boxes) and semantic prompts (text) in the problem formulation. Since then, Text vs. Layout Trade-off would be justified!
> >
> > * ,3. **The claim that “this effectively empowers users to implicitly control the pose via the box shape” is not sufficiently convincing**. In practice, adjusting the bounding box aspect ratio to induce a desired pose -- without any explicit pose or instance-level guidance -- amounts to a trial-and-error process rather than a principled control mechanism. Moreover, if the text prompt specifies a pose that contradicts the geometric implication of the bounding box, it is unclear how the model resolves this conflict during generation. The current formulation does not appear to include any internal mechanism, such as conflict arbitration, prompt re-weighting, or pre-processing, to reconcile inconsistencies between semantic (text) and geometric (box) cues.
> >
> > * 4-5. The authors' claims address my concern.

---

> ### Author Response · Authors · 2026-01-10
>
> We thank the reviewers for their detailed feedback and have updated the submission to address the raised concerns.
>
> 1–2. Problem Formulation: Roles of Box vs. Text Prompts
> We thank the reviewer for this crucial clarification. In the revised manuscript , we have formalized the problem definition to explicitly distinguish the roles of spatial and semantic prompts under a "Layout-Priority" paradigm:
>
> • **Bounding boxes** function as **hard spatial constraints** (defining the *mandatory* generation region).
>
> • **Text prompts** function as **semantic references** (defining the *content* within those regions).
> We explicitly acknowledge that conflicts (e.g., text says “large” vs. a small box) are unavoidable. Rather than assuming an implicit conflict-aware training, our architecture structurally resolves such conflicts by **prioritizing the box**. Since our *Subject-Grounded Cross-Attention* physically masks the attention maps, the model is forced to satisfy the geometric constraint first.
>
> We have added such definition in **Section 3, page3(marked red)** in the revised version to discuss this text–layout trade-off as an inherent design choice rather than a modeling deficiency.
>
>
> 3. Box Shape, Pose Control, and Conflicts
> We appreciate the correction. We agree that adjusting aspect ratios provides "coarse geometric guidance" rather than principled pose control. We have revised the manuscript to remove claims of explicit control.
>
> • **Conflict Resolution:** When semantic cues in the text contradict geometric cues implied by the bounding box, our formulation explicitly prioritizes the box as a hard spatial constraint. The model generates content strictly within the specified region and does not attempt to reconcile or arbitrate conflicting semantic and spatial signals. As a result, geometric properties of the box may influence the generated appearance in an emergent and non-deterministic manner. We clarify in the Limitations (supplementary section J, marked in red) that this behavior should be viewed as a side effect of spatial constraint rather than a principled control mechanism, and that no internal conflict arbitration or prompt re-weighting is employed.
>
> We explicit clarify that no internal conflict arbitration is assumed in the current system. However, we agree that integrating an LLM-based layout generator (e.g., LayoutGPT ) to pre-validate or refine user inputs would be a promising strategy to resolve such semantic-geometric conflicts automatically in future work.” We have added this into the Limitations (supplementary section J, marked in red)
>
> 6. Motivation for Fourier Positional Encoding (FPE)
> We agree the motivation should be theoretically grounded. In the revision, we justify FPE based on the "Spectral Bias" of neural networks [Fourier Features Let Networks Learn High Frequency Functions in Low Dimensional Domains, Tancik et al., NeurIPS 2020].
>
> • **Rationale:** Grounding tasks involve learning sharp boundaries (i.e., the transition from "subject" to "background" at the box edge is effectively a high-frequency step function). Standard normalized coordinates induce a low-frequency bias, making it difficult for MLPs to fit these sharp transitions, leading to "leaking" attention. FPE maps coordinates to a higher-dimensional space, enabling the model to capture these **high-frequency spatial constraints** precisely. We have added corresponding ablation study in the revised version of submission(section 4.6 and table 7, marked red). Ablation results demonstrate that substituting Fourier encoding with linear encoding leads to a noticeable degradation in layout alignment, as reflected by a decline in the YOLO-AP score. This confirms that Fourier Positional Encoding is essential for mapping positional information into a high-frequency feature space, thereby overcoming the spectral bias of neural networks and enabling the model to strictly adhere to bounding box constraints.
>
> 7. Random Padding, SAM-2 Noise, and Preprocessing
>
> Thank you for highlighting these implementation details. We have expanded the Implementation Details section to clarify our robustness strategies:
>
> To prepare the training data, we utilize SAM-2 to extract object masks. To ensure data quality, we have already filtered out small objects occupying less than 5\% of the image area and larger than 80 % area of the image area.
> Crucially, to preserve local semantic context around the subject, we apply a random padding strategy during cropping, with a scaling factor $s$ sampled uniformly from $[1.1, 1.3]$. Regarding segmentation noise, we observe that minor imperfections in SAM-2 masks effectively serve as a form of data augmentation. This noise injection during training enhances the model's robustness, enabling it to better handle imprecise, user-drawn bounding boxes during the inference phase.

---

### Review · Reviewer_C87i · 2025-12-21

**Summary Of Contributions:**

This paper proposes GroundingBooth, a framework for grounded text-to-image customization. The task aims to generate images where: (1) foreground subjects preserve the identity of input reference images, (2) both subjects and background text entities adhere to specified bounding box layouts, and (3) the overall image aligns with the text prompt.

The technical contributions include:
* A subject-grounded cross-attention layer that uses layout masks to restrict the injection of subject features within corresponding bounding boxes, preventing context blending between foreground and background.
* The framework supports multi-subject customization at inference time despite being trained on single-subject data.

Strengths:
* The task formulation of jointly grounding both subject-driven foreground and text-driven background is practically relevant.
* Comprehensive experiments on DreamBench and COCO with multiple evaluation metrics (CLIP-T, CLIP-I, DINO, AP, FID).
* The paper is clearly written and the method is easy to follow.

Weaknesses:
* Limited technical novelty: the gated self-attention is directly from GLIGEN, and the subject-grounded cross-attention is similar to existing masked attention mechanisms in prior works.
* Generated subjects exhibit nearly identical poses to input reference images, contradicting claims of generating "natural poses."
* The distinction from MS-Diffusion (already accepted at ICLR 2025) is not convincingly demonstrated.

**Audience:**

Yes

**Audience Explanation:**

The problem of jointly controlling layout and preserving subject identity in text-to-image generation is practically important for applications such as  E-commerce product visualization and controllable image editing.

**Broader Impact Concerns:**

No major ethical concerns are identified with this work. The method focuses on personalized image generation with layout control, which has applications in creative tools and content creation. As with all subject-driven generation methods, there is potential for misuse in generating misleading imagery of objects/products. However, this is a general concern for the field rather than specific to this work.

**Claims And Evidence:**

No

**Claims Explanation:**

1. The claimed novelty of the subject-grounded cross-attention layer is limited.The paper presents this as a key contribution, but the core technique of using layout masks to constrain cross-attention regions (Eq.4-5) has been explored in prior works:
* "Be Yourself: Bounded Attention for Multi-Subject Text-to-Image Generation" (ECCV 2024)
* InstanceDiffusion (CVPR 2024) uses masked attention for instance-level control
* "Training-Free Layout Control with Cross-Attention Guidance" (WACV 2024)

2. The distinction from MS-Diffusion is not convincingly supported. The paper claims MS-Diffusion "fails to spatially control the background contents" (Sec 2). However:
* Table 1 shows MS-Diffusion achieves higher CLIP-T (0.3029) than GroundingBooth (0.2931), suggesting better text alignment including background descriptions.
* No direct experiment demonstrates GroundingBooth's superior background grounding capability.

3. The claim of generating subjects with "natural poses" (Sec 3) is not supported by visual evidence. In Fig.1, Fig.5, and Fig.6, the generated subjects consistently maintain nearly identical poses to their input reference images.

**Requested Changes:**

1. Clarify novelty over prior masked attention methods. Provide detailed technical comparison with Be Yourself (ECCV 2024), InstanceDiffusion (CVPR 2024), and Training-Free Layout Control (WACV 2024). Explain what is fundamentally new in the subject-grounded cross-attention beyond applying layout masks to attention.

2. Provide convincing evidence for background grounding superiority over MS-Diffusion. Include:
    * Direct comparison on background object placement accuracy
    * Ablation showing that background text entities are correctly grounded
    * Update the "concurrent work" framing since MS-Diffusion predates this submission
3. Address the pose consistency issue. Provide examples where generated subjects exhibit pose variations adapted to the scene context, or explicitly acknowledge this as a limitation. Currently, the visual results contradict the claim of generating "natural poses."

---

> ### Author Response · Authors · 2026-01-02
>
> We sincerely thank the reviewer for the constructive feedback and for recognizing the practical relevance of our task, the clarity of our writing, and the comprehensiveness of our experiments. We address the specific concerns regarding novelty, the comparison with MS-Diffusion, and pose consistency below.
>
>
> **1. Clarification on Novelty and Distinction from Prior Works**
>
> Regarding the novelty of our masked attention mechanism compared to prior works (e.g., *InstanceDiffusion*, *Be Yourself*), we clarify that our **Subject-Grounded Cross-Attention (SG-CA)** is designed for a fundamentally different challenge: **encoder-based customization**. Unlike prior methods that operate in homogeneous text-to-image settings, our task involves **heterogeneous conditioning** where information-dense identity tokens from the reference image compete with semantic text tokens. Without explicit disentanglement, visual features often "leak" into the background. Our SG-CA physically isolates the receptive field of identity tokens to specific regions, serving as a **disentanglement module** rather than just a spatial constraint. This ensures the background is generated strictly from text entities while the subject preserves high-fidelity identity, achieving superior DINO scores (0.7950) compared to baselines. We will add a detailed discussion on these technical distinctions in the related work section.
>
> **2. Distinction from MS-Diffusion and Background Grounding Evidence**
>
> Concerning the comparison with MS-Diffusion, we emphasize two key distinctions. First, Grounding Booth uniquely supports **explicit background grounding** via bounding boxes for text entities (e.g., "mountain", "lake"), whereas MS-Diffusion primarily focuses on subject layout and lacks this granular control over background text entities. MS-Diffusion naturally does not support the grounded generation of background text entities. We have already added additional qualitative comparison in Fig.9 of the revised version of submission to illustrate this difference.
>
> Second, regarding the lower CLIP-T score, we attribute this to MS-Diffusion's use of the larger **SDXL backbone**, which has stronger native text alignment than the SD v1.4 base used in our main experiments. We regard adapting our model to more advanced base model such FLUX, FLUX2 as our future work. However, despite using a smaller backbone, GroundingBooth significantly outperforms MS-Diffusion in **Identity Preservation (DINO: 0.7950 vs. 0.7267)** , demonstrating that our architecture is more effective at the core task of preserving subject details. We also add VLM-based methrics in Table 6 in the revised version of submission to further measure the text alignment ability of the model, and our method shows competitive performance in prompt-following ability.
>
> For the background grounding ability, we have already shown sufficient examples in Fig.1, Fig.6, Fig.7, and supplementary Fig.5. Specifically in Fig.7 and supplementary Fig.5, we test our model on COCO dataset. Even in complex scenes with more than 8 background text entities, our method still achieve precise grounded generation of background text entities. This has already demonstrated the background grounding ability of our proposed method.
>
>
> We have already removed the “concurrent work” in the revised version of submission.
>
> **3. Clarification on "Natural Poses"**
>
> We respectfully refer the reviewer to **Supplementary Figures 1 and 4**, which illustrate examples where pose adaptations do occur. We wish to clarify that in our grounded customization task, the pose of the generated foreground is **jointly constrained** by two factors: (1) the **aspect ratio and dimensions** of the reference object's bounding box, and (2) the model’s objective to **harmonize** the subject with the background context. Since our method enforces strict spatial control, the potential for pose variation is physically limited by the user-provided layout. For instance, a narrow bounding box (as seen in the "boot" example in Fig. 5, Row 2 ) leaves limited spatial freedom for the subject to shift its pose. Therefore, the observed consistency often reflects the model’s adherence to these precise spatial constraints, rather than an inherent inability to generate pose variations.

---

### Decision · Action_Editor_LWRC · 2026-02-04

**Recommendation:** Accept with minor revision

**Additional Comments:**

I think this paper is borderline. In particular, Reviewer C87i has two specific concerns, which are quoted below:

(1) "**Pose generation claim.** I observed that generated poses largely follow the input references, which contradicts the claim of generating "natural poses." Reviewer xcvP raised a related concern that using bounding box geometry to influence pose is "trial-and-error rather than a principled control mechanism." The authors acknowledged that pose is constrained by bounding box geometry."

Please clearly acknowledge this limitation in the final submission.

(2) "**Comparison with MS-Diffusion.** I questioned whether the claimed advantage over MS-Diffusion is convincing, given that MS-Diffusion achieves higher CLIP-T (0.3029 vs 0.2931). The authors attributed this gap to backbone differences (MS-Diffusion uses SDXL while this work uses SD v1.4). However, no experiments on SDXL were added to enable a fair comparison, leaving the claimed superiority over MS-Diffusion insufficiently supported."

Please add experiments to enable a fair comparison, as instructed above. If for any reason this is intractable, please respond and explain.

(3) Please make all the modifications/revisions promised below in the author responses.

**Audience:**

Yes

**Audience Explanation:**

My understanding is that the computer vision community will be interested in this paper.

**Claims And Evidence:**

Yes

**Claims Explanation:**

This is a borderline paper. Two out of three reviewers think that the claims made in the submission are supported by accurate, convincing, and clear evidence, but Reviewer C87i disagrees.

Specifically, Reviewer C87i  points out that the claimed advantage over MS-Diffusion might not be convincing. I agree with Reviewer C87i's point technically, but I think this is a relatively small part of the paper. My understanding is that most claims in this submission are well supported by accurate, convincing, and clear evidence, and I would give Yes to this question.